# Stereoselective synthesis of medium lactams enabled by metal-free hydroalkoxylation/ stereospecific [1,3]-rearrangement

Bo Zhou[1], Ying-Qi Zhang[1], Kairui Zhang[2], Ming-Yang Yang[1], Yang-Bo Chen[1], You Li[2], Qian Peng [2], Shou-Fei Zhu [2], Qi-Lin Zhou[2] & Long-Wu Ye [1,3]

Rearrangement reactions have attracted considerable interest over the past decades due to their high bond-forming efficiency and atom economy in the construction of complex organic architectures. In contrast to the well-established [3,3]-rearrangement, [1,3] O-to-C rearrangement has been far less vigorously investigated, and stereospecific [1,3]-rearrangement is extremely rare. Here, we report a metal-free intramolecular hydroalkoxylation/ [1,3]-rearrangement, leading to the practical and atom-economical assembly of various valuable medium-sized lactams with wide substrate scope and excellent diastereoselectivity. Moreover, such an asymmetric cascade cyclization has also been realized by chiral Brønsted acid-catalyzed kinetic resolution. In addition, biological tests reveal that some of these medium-sized lactams displayed their bioactivity as antitumor agents against melanoma cells, esophageal cancer cells and breast cancer cells. A mechanistic rationale for the reaction is further supported by control experiments and theoretical calculations.

[1] State Key Laboratory of Physical Chemistry of Solid Surfaces, Key Laboratory of Chemical Biology of Fujian Province, and College of Chemistry and Chemical Engineering, Xiamen University, 361005 Xiamen, China. [2] State Key Laboratory and Institute of Elemento-Organic Chemistry, College of Chemistry, Nankai University, 300071 Tianjin, China. [3] State Key Laboratory of Organometallic Chemistry, Shanghai Institute of Organic Chemistry, Chinese Academy of Sciences, 200032 Shanghai, China. Correspondence and requests for materials should be addressed to Q.P. (email: qpeng@nankai.edu.cn) or to S.-F.Z. (email: sfzhu@nankai.edu.cn) or to L.-W.Y. (email: longwuye@xmu.edu.cn)

Eight-membered lactams, especially the benzo[*d*]azocinones, are prominent structural motifs that can be found in many natural products and bioactive molecules (Fig. 1)[1–5]. However, access to these heterocycles is challenging owing to unfavorable enthalpic and entropic barriers in transition states leading to medium-sized rings[6–10]. To date, only very limited methods have been developed, and most of them rely on noble-metal catalysis[11–19]. To this end, the development of new methods for the efficient construction of this skeleton is highly desirable, especially those with high diastereo- and enantioselectivity.

Rearrangement reactions have attracted considerable interest over the past decades due to their high bond-forming efficiency and atom economy in the construction of complex organic architectures[20,21]. In contrast to the well-established [3,3]-rearrangement[22–25], which generally proceeds via the chair-like transition state and thus is stereospecific (Figs. 2a), [1,3] O-to-C rearrangement has been far less vigorously investigated, and stereospecific [1,3]-rearrangement is highly challenging due to the formation of presumable zwitterion pairs (Fig. 2b)[26,27]. Although several Lewis acid-mediated and thermal [1,3]-rearrangements that relay stereochemical information have been reported[28,29], transformation in these limited cases lacks generality and significant deterioration of enantiomeric excess is observed[30,31].

Recently, great progress of transition metal-catalyzed intramolecular alkoxylation-initiated [1,3]-rearrangement has evoked a new round of exploration on the [1,3]-rearrangements, offering great potential to build structurally complex cyclic molecules, as elegantly established by Toste, Rhee, Hashmi, Liu, Davies, and Zhu[31–38]. Despite these impressive advances, these tandem reactions are limited to ether nucleophiles and rely on noble metals (Au/Pt) as the catalyst. Importantly, no direct catalytic asymmetric tandem reaction has been described to date[39]. Inspired by the above results and by our recent study on yttrium-catalyzed tandem intramolecular hydroalkoxylation/Claisen rearrangement[40], we envisioned that the synthesis of eight-membered benzo[*d*]azocinones **2** might be accessed directly through catalytic intramolecular hydroalkoxylation/[1,3]-rearrangement of ynamides **1**[41–50]. Herein, we describe the realization of a metal-free tandem intramolecular hydroalkoxylation/[1,3]-rearrangement (Fig. 2c), and this method leads to the practical and atom-economical synthesis of various valuable medium-sized lactams with excellent diastereoselectivity. Moreover, such an asymmetric cascade cyclization has also been achieved via kinetic

resolution by chiral spiro phosphoramide catalysis. Importantly, this [1,3]-rearrangement is highly stereospecific and proceeds with complete chirality transfer. Control experiments and density functional theory (DFT) calculations provide further evidence of the feasibility of the proposed mechanism.

## Results

**Screening of reaction conditions.** At the outset, ynamide **1a** was used as the model substrate to demonstrate our designed cascade cyclization, as shown in Table 1 (for more details see Supplementary Table 1). To our delight, the expected benzo[*d*]azocinone **2a** was indeed formed with exclusive *cis*-diastereoselectivity (diastereoselectivity (d.r.) >50:1; determined by crude proton nuclear magnetic resonance (¹H NMR)), albeit in low yields, in the presence of typical gold catalysts (Table 1, entries 1 and 2). Somewhat surprisingly, further investigations revealed that the reaction also proceeded in the presence of various non-noble metals (Table 1, entries 4–7), with Zn(OTf)$_2$ giving the best yield of the desired product **2a** (Table 1, entry 7). In addition, Brønsted acids such as TsOH and MsOH could also catalyze this cascade reaction to produce **2a** in 47 and 66% yields, respectively, together with significant amounts of hydration product **2a′** in both cases (Table 1, entries 8 and 9). Although the use of 10 mol% of HOTf as catalyst failed to produce the desired **2a**, probably because the high acidity led to decomposition of **1a** (Table 1, entry 10), the reaction efficiency was substantially improved by decreasing the loading of catalyst (Table 1, entries 11–13). With a low catalyst loading of 0.5 mol%, HOTf efficiently catalyzed the formation of **2a** in 96% yield (Table 1, entry 13). These results indicate that HOTf, which was released as a hidden Brønsted acid, is presumably the true catalytic species in the above Lewis acid catalysis[51–53].

**Reaction scope study.** The reaction scope was then explored under the optimized reaction conditions (Fig. 3). This metal-free tandem reaction occurred efficiently with various benzyl alcohol-tethered ynamides **1**, leading to the corresponding benzo[*d*]azocinones **2** in good to excellent yields. Importantly, excellent diastereoselectivity (>50:1) was achieved in all cases. Ynamides with different sulfonyl-protecting groups were first investigated, and the desired products **2a–2c** were formed in 73–94% yields. In addition, ynamides bearing either electron-withdrawing or electron-donating substituents, such as F, Cl, Br, Me, OMe, or

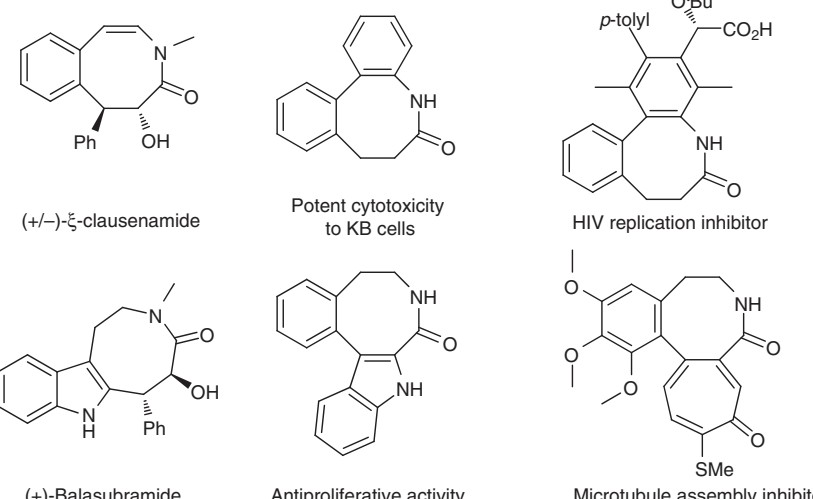

**Fig. 1** Benzo[*d*]azocinones in natural products and bioactive molecules. Some of representative molecules are listed

(+/–)-ξ-clausenamide

Potent cytotoxicity to KB cells

HIV replication inhibitor

(+)-Balasubramide

Antiproliferative activity

Microtubule assembly inhibitor

**Fig. 2** [3,3]-Rearrangement vs. [1,3]-rearrangement. **a** Typical [3,3]-rearrangement. **b** Typical [1,3]-rearrangement. **c** This work: Brønsted acid-catalyzed hydroalkoxylation/stereospecific [1,3]-rearrangement

**Table 1 Optimization of reaction conditions[a]**

| Entry | Catalyst | Yield (%)[b] | |
|---|---|---|---|
| | | 2a | 2a' |
| 1 | IPrAuNTf₂ (5 mol%) | 48 | 3 |
| 2 | Ph₃PAuNTf₂ (5 mol%) | 32 | <1 |
| 3 | AgOTf (10 mol%) | 15 | <1 |
| 4 | Cu(OTf)₂ (10 mol%) | 18 | 5 |
| 5 | Y(OTf)₃ (10 mol%) | 73 | 4 |
| 6 | Yb(OTf)₃ (10 mol%) | 74 | 3 |
| 7 | Zn(OTf)₂ (10 mol%) | 76 | 3 |
| 8 | TsOH (10 mol%) | 47 | 12 |
| 9 | MsOH (10 mol%) | 66 | 10 |
| 10 | HOTf (10 mol%) | <5 | <1 |
| 11 | HOTf (5 mol%) | 72 | <1 |
| 12 | HOTf (1 mol%) | 88 | <1 |
| **13** | **HOTf (0.5 mol%)** | **96** | **<1** |

[1]H NMR proton nuclear magnetic resonance
[a]Reaction conditions: **1a** (0.1 mmol), catalyst (0.5–10 mol%), PhCl (2 mL), 80 °C, 4 h, in vials
[b]Measured by [1]H NMR using diethyl phthalate as internal standard

even CN and CF₃ on the aromatic ring (R² = Ar), were compatible with this cyclization to produce the expected **2d–2l** in generally excellent yields. This cascade cyclization was also extended to the naphthalene, thiophene, and alkenyl-substituted

ynamides, delivering the desired **2m** (99%), **2n** (81%), and **2o** (94%), respectively. Various aryl-substituted ynamides with either electron-donating or electron-withdrawing groups were then screened, and the reaction afforded the desired products **2p–2ab**

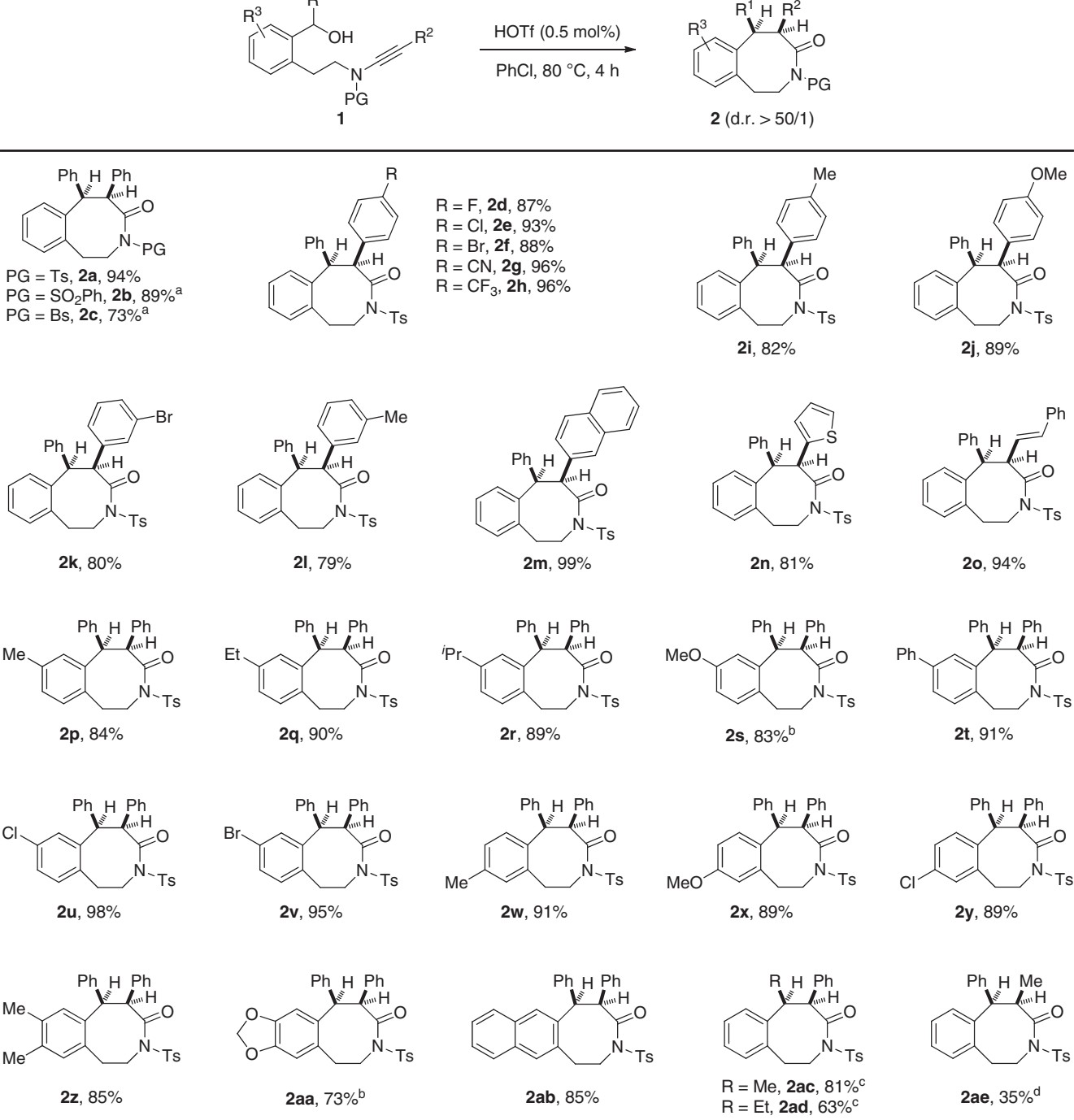

**Fig. 3** Reaction scope for the formation of 3-benzazocinones **2**. Reaction conditions: **1** (0.2 mmol), HOTf (0.001 mmol), PhCl (4 mL), 80 °C, 4 h, in vials; yields are those for the isolated products. [a]1 mol% of HOTf was used. [b]Using 10 mol% of Zn(OTf)$_2$ as catalyst and 5 Å molecular sieve (MS) as additive. [c]100 °C, 60 h. [d]100 °C, 4 h

in 73–98% yields. Of note, in some cases better yields could be achieved by employing Zn(OTf)$_2$ (10 mol%) as catalyst and 5 Å molecular sieve (MS) as additive (**2s** and **2aa**). Interestingly, alkyl-substituted ynamides (R$^1$ or R$^2$ = alkyl) were also suitable substrates, and were converted into the desired **2ac** and **2ad** in good yields, and **2ae** in a serviceable yield; higher temperature was needed in these cases. The molecular structures of **2a** and **2ac** were confirmed by X-ray diffraction (for more details see Supplementary Tables 3 and 4).

Notably, this cascade cyclization was also extended to the allyl alcohol-tethered ynamides, and, importantly, no competing intramolecular hydroalkoxylation/[3,3]-rearrangement was observed[40]. As shown in Fig. 4a, the desired benzo[d]azocinones **2af–2ah** were obtained in 61–76% yields, and significantly improved yield (86%) was achieved in case of ynamide **1ah** by using Zn(OTf)$_2$ as catalyst. In addition, the reaction proceeded smoothly to produce the expected 9-membered lactam **2ai** in 46% yield, and, in this case, the use of Zn(OTf)$_2$ as catalyst also gave significantly improved yield

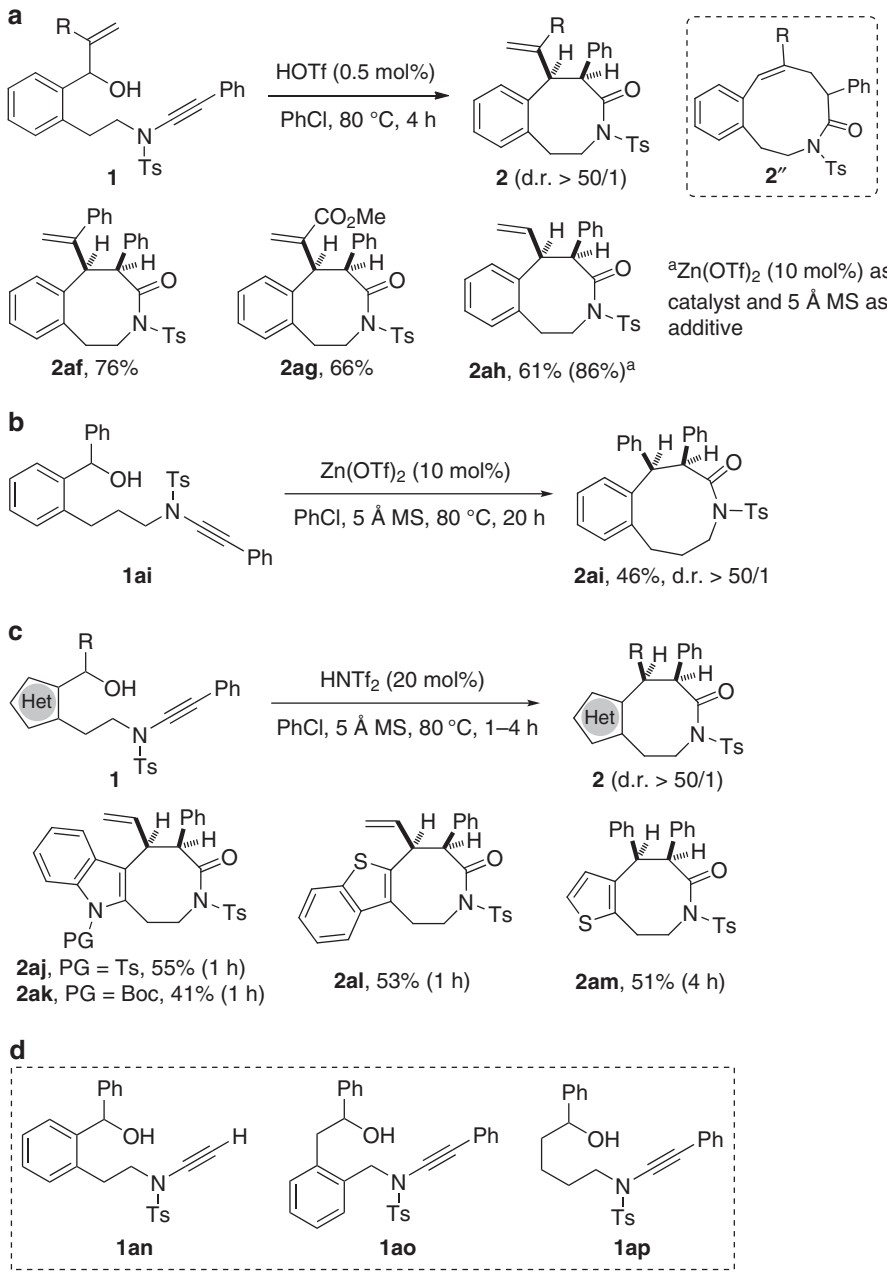

**Fig. 4** Catalytic hydroalkoxylation/[1,3]-rearrangement of other ynamides **1**. **a** Brønsted acid-catalyzed hydroalkoxylation/[1,3]-rearrangement of allyl alcohol-tethered ynamides **1af**–**1ah**. **b** Zinc-catalyzed cascade cyclization of ynamide **1ai**. **c** HNTf$_2$-catalyzed cascade cyclization of other heterocycle-linked ynamides **1aj**–**1am**. **d** Ynamides **1an**–**1ap** which failed to give the desired products

(Fig. 4b, **2ai** was confirmed by X-ray diffraction, for more details see Supplementary Table 5). Moreover, it was found that other heterocycle-linked 8-membered ring lactams **2aj**–**2am** could also be synthesized in 41−55% yields in the presence of 20 mol% of HNTf$_2$ as catalyst (Fig. 4c). Attempts to extend the reaction to the terminal ynamide **1an** only gave a complex mixture of products, and the reaction of ynamides **1ao** and **1ap** also failed to produce the desired products (for more details see Supplementary Figs. 118 and 119), indicating that the formation of a stable benzylic carbocation is a key requirement for subsequent [1,3]-rearrangement (Fig. 4d).

**Screening of reaction conditions for kinetic resolution.** We then considered the possibility of developing an asymmetric variant of this tandem sequence. Although no enantioselectivity was observed by the use of chiral metal catalysts, good enantioselectivity

could be attained by employing chiral spiro phosphoramides as catalysts (for more details, see Supplementary Table 2 and Supplementary Fig. 120)[54,55]. Importantly, further studies revealed that the chiral induction was realized through kinetic resolution of racemic ynamides (for more details, see Supplementary Fig. 121). Initially, ynamide **1p**, bearing an electron-donating methyl group on the aromatic ring moiety that should promote this cascade cyclization, was used as the model substrate. As shown in Table 2, the desired chiral benzo[*d*]azocinone **2p**-*ent* was obtained in 42% yield with an enantiomeric ratio (e.r.) of 95:5 in the presence of chiral spiro phosphoramide **Cat. 3**[56–59], bearing two 6,6'-di(3,5-di-*tert*-butyl-4-methoxyphenyl) moieties (Table 2, entry 5). Interestingly, the use of the corresponding chiral binol-derived phosphoramide led to significantly decreased enantioselectivity (e.r. <60:40), indicating that the spirobiindane backbone of the phosphoramides plays a crucial

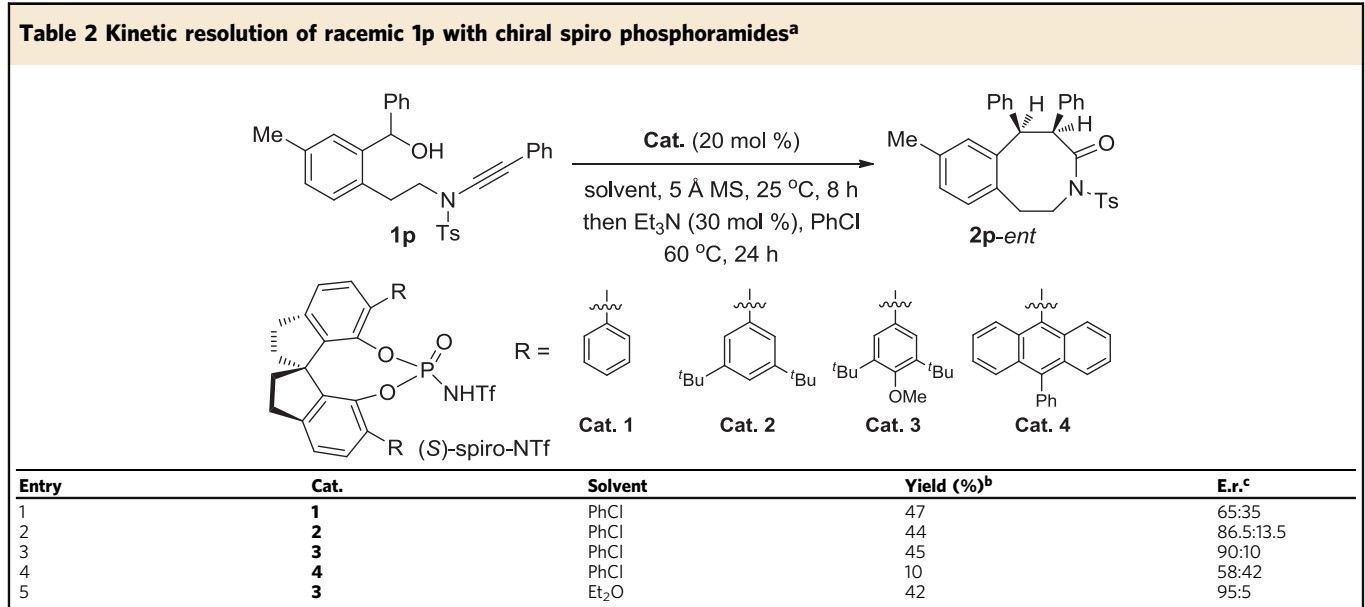

**Table 2 Kinetic resolution of racemic 1p with chiral spiro phosphoramides[a]**

| Entry | Cat. | Solvent | Yield (%)[b] | E.r.[c] |
|---|---|---|---|---|
| 1 | 1 | PhCl | 47 | 65:35 |
| 2 | 2 | PhCl | 44 | 86.5:13.5 |
| 3 | 3 | PhCl | 45 | 90:10 |
| 4 | 4 | PhCl | 10 | 58:42 |
| 5 | 3 | Et₂O | 42 | 95:5 |

*HPLC* high-performance liquid chromatography
[a]Reaction conditions: **1p** (0.1 mmol), **Cat**. (0.02 mmol), solvent (2 mL), 25 h, 8 h, then Et₃N (0.03 mmol), PhCl (1 mL), 60 °C, 24 h, in vials
[b]Isolated yields
[c]Determined by HPLC analysis on a chiral stationary phase

**Fig. 5** Reaction scope for kinetic resolution of racemic **1**. Reaction conditions: **1** (0.1 mmol), **Cat. 3** (0.02 mmol), Et₂O (2 mL), 25 °C, 6–32 h, then Et₃N (0.03 mmol), PhCl (1 mL), 60 °C, 24 h, in vials; yields are those for the isolated products; e.r.s are determined by high-performance liquid chromatography (HPLC) analysis on a chiral stationary phase

role in the chiral induction step. It is notable that in this process one enantiomer ((*R*)-**1p**) favored formation of the desired chiral benzo [*d*]azocinone **2p**-*ent*, while the other enantiomer ((*S*)-**1p**), which does not match with the **Cat. 3**, favored formation of the corresponding hydration product **2p′** catalyzed by the acid (for more details, see Supplementary Figs. 122–124). Thus, it represents a rare example of parallel kinetic resolution[60,61].

**Scope of kinetic resolution of racemic ynamides 1.** Preliminary investigations were carried out into the reaction scope by employing chiral spiro phosphoramide **Cat. 3** as a catalyst (Fig. 5). Substrates with either electron-donating or electron-withdrawing groups on the aromatic ring moiety of the racemic ynamides **1** were well tolerated and resulted in 40–51% yields and good e.r. values. The absolute configuration of **2p**-*ent* was

**Fig. 6** Gram-scale reaction and product elaboration. Gram-scale reaction of ynamide **1a** and transformation of **2a** into **3a**, **4a**, and **5a**

**Fig. 7** Control experiments. **a** Cascade cyclization of ynamide **1a** by quenching the reaction after 15 min. **b** Control experiments on the transformation of racemic **6a** into racemic **2a**. **c** Control experiments on the transformation of chiral **6a** into chiral **2a**

determined by X-ray crystallographic analysis (for more details, see Supplementary Table 6). Of note, although 20 mol% catalyst loading was employed, probably because the acidity of the chiral catalyst is not high enough, the catalyst could be readily recovered and reused five times with almost unchanged enantioselectivity and reactivity (for more details, see Supplementary Fig. 125).

**Synthetic applications and biological tests**. To further demonstrate the potential utility of this reaction, we also carried out product derivatizations (Fig. 6). For example, the Ts group in benzo[*d*]azocinone **2a**, prepared on a gram scale in 86% yield, was efficiently removed to form free amide **3a** in 87% yield by the treatment with SmI$_2$. **3a** could be further methylated into the corresponding lactam **4a** (89%) and oxidized into unsaturated lactam **5a** (72%), respectively.

Moreover, we tested the above-synthesized 3-benzazocinones for their bioactivity as antitumor agents. The cytotoxic effects of these compounds were evaluated against a panel of cancer cells, including melanoma cells A375, esophageal cancer cells SK-GT-4 and KYSE-450, and breast cancer cells MCF-7 and MDA-MB-231 using cell viability assay. Our preliminary studies showed that almost half of these compounds exerted significant cytotoxic effects on the A375, and a few compounds (**2ac**, **2ah**, **2p**-*ent*, and **3a**) and compound **2am** exerted cytotoxic effects on the SK-GT-4 and MCF-7 (for more details, see Supplementary Table 8),

respectively, suggesting a potential application of these medium-sized lactams in medicinal chemistry.

**Mechanistic investigations**. We then turned our attention to mechanistic investigations (for more details, see Supplementary Figs. 126–129). First, it was found that no incorporation of $^{18}$O into the product **2a** was observed when ynamide **1a** was subjected to the reaction conditions with H$_2$$^{18}$O, which indicates that the oxygen on the carbonyl group of **2a** originates from the hydroxyl group of **1a** (for more details, see Supplementary Fig. 126). In addition, hydration product **2a′** was not converted into **2a** under the standard conditions, thus ruling out **2a′** as a possible intermediate (for more details, see Supplementary Fig. 127). Gratifyingly, the ketene aminal **6a** (only the *E* isomer) could be isolated in 53% yield by quenching the reaction after 15 min (Fig. 7a). Importantly, **6a** was readily converted into the desired **2a** and complete chirality transfer was observed starting from chiral **6a** (Fig. 7b, c). Furthermore, the acid catalyst did not work in this rearrangement process, indicating that it is an uncatalyzed thermal rearrangement (for more details, see Supplementary Fig. 128). These results strongly support that **6a** is the key intermediate and stereospecific [1,3]-rearrangement is presumably involved in this tandem process.

On the basis of the above observations, we propose a mechanism for the formation of benzo[*d*]azocinone **2a** (Fig. 8). The reaction

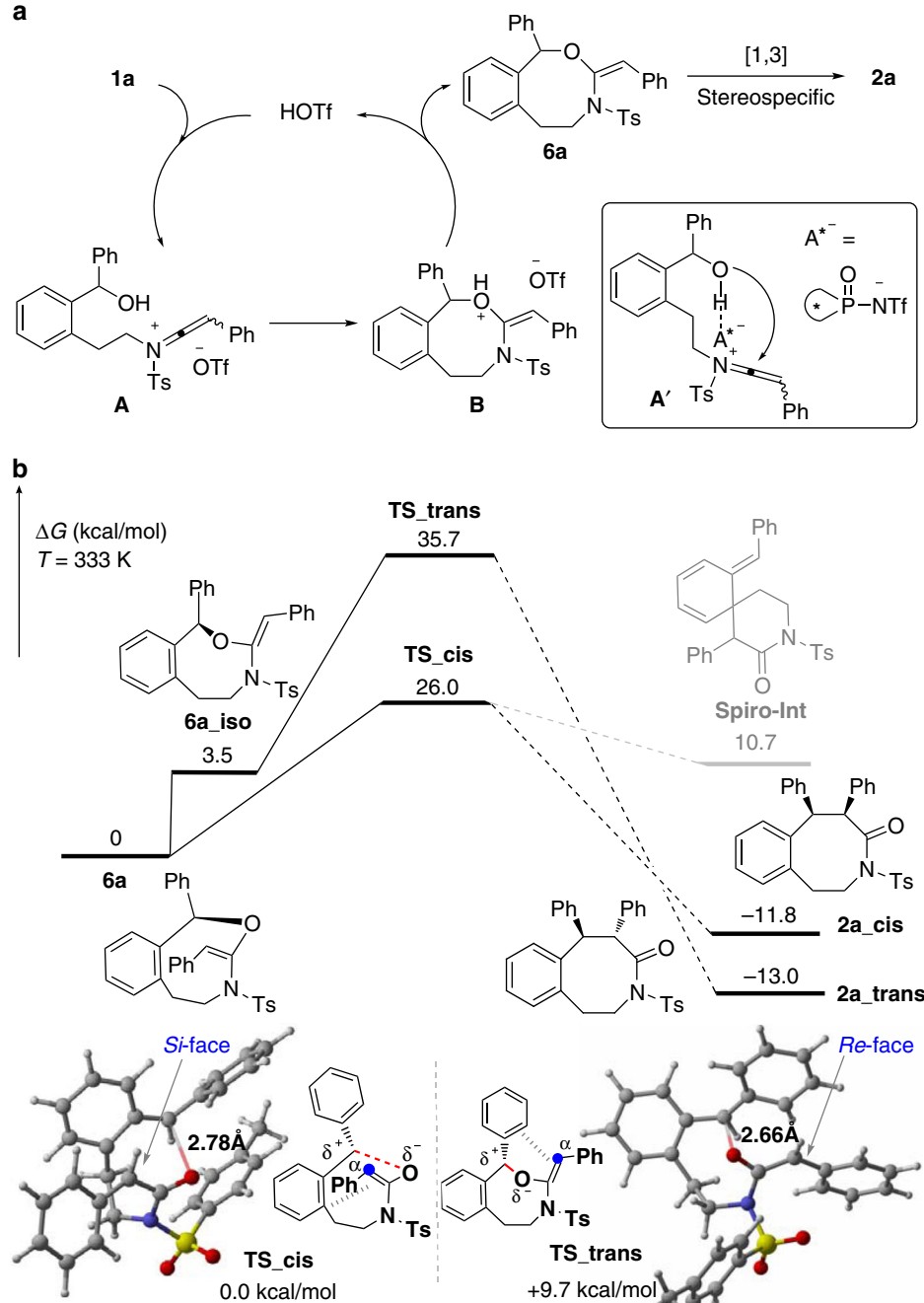

**Fig. 8** Mechanistic hypothesis. **a** Plausible catalytic cycle. **b** Density functional theory (DFT) calculations on the O-to-C rearrangement of (*R*)-**6a**

begins with the hydroxyl group attack of the HOTf-activated ynamide **1a** to afford oxonium intermediate **B** presumably via a keteniminium intermediate **A**, thus further yielding thermodynamically stable vinyl ether intermediate **6a** of *E* configuration and regenerating the acid catalyst. DFT calculations at B3LYP-D3/6-31G(d,p)//SMD-def2-TZVP level of theory[62–65] (for more details, see Supplementary Figs. 111–113 and Supplementary Datasets 1–4) were introduced to understand the subsequent O-to-C rearrangement of (*R*)-**6a**, which is a stereospecific, rate-determining, and uncatalyzed thermal rearrangement, to produce the final *cis* product (*R*, *S*)-**2a**. Two major conformational isomers of initial compound **6a** and **6a-iso** can be located by calculations in terms of different *Si*- or *Re*-face at the α-carbon of amide substrate. In the following step of C-O bond cleavage, transition state **TS_cis** with bond breaking

via *Si*-face side was 9.7 kcal/mol more stable than that via *Re*-face cleavage-formed **TS_trans**, suggesting the final **2a_cis** from **TS_cis** is kinetically favorable and highly stereospecific. Furthermore, calculations indicate that the reaction mechanism is not a typical [1,3]-rearrangement (for more details, see Supplementary Figs. 114–117)[66–70]. We are able to locate the transition states of C-O bond cleavage, but failed to locate transition states of C-C bond formation, which indicate that the mechanism seems not to be stepwise. Quantitative Sensory Testing method was used and three levels of methods (B3LYP-D3, M062X, ωB97XD) were tried in our calculations. All of them give substantially similar kinetic and thermodynamic results (for more details, see Supplementary Figs. 111–117). However, more inspections for transition states imply that the developing negative charge on the α-carbon of amide

substrate tends to be stabilized by aromatic rings forming formal [3,3]-rearrangement transition states. A regular spiro[5,5] product **spiro-int** is not accessible because of its high energy instability for 22.5 kcal/mol, compared to the energy of **2a-cis**. Indeed, transition states of C–C bond formation are facile to the final product during our calculations. Based on energy profiles, the pathway to form favorable **2a-cis** is kinetically irreversible and thermodynamically exothermic for 11.8 kcal/mol, which is in agreement with our experiment. When chiral Brønsted acid is employed, the resulting keteniminium intermediate **A′** leads to chiral **6a** via ion pairing and H-bonding interactions[54,55], which undergoes stereospecific [1,3]-rearrangement to form **2a**-ent with complete chirality transfer.

## Discussion

In summary, we have achieved a metal-free intramolecular hydroalkoxylation/[1,3]-rearrangement, and significantly, this [1,3]-rearrangement is highly stereospecific, and a mechanistic rationale for this stereospecificity is also strongly supported by DFT calculation. This method leads to the practical and atom-economical synthesis of a diverse array of valuable medium-sized lactams from readily available ynamides in high yields with broad substrate scope and excellent diastereoselectivity. Furthermore, this asymmetric cascade cyclization has also been realized via kinetic resolution by chiral spiro phosphoramide catalysis, thus constituting a rare example of chiral Brønsted acid-catalyzed kinetic resolution. In addition, biological tests reveal that some of these medium-sized lactams displayed their bioactivity as anti-tumor agents against melanoma cells, esophageal cancer cells, and breast cancer cells. We anticipate that the mechanistic insights of this chemistry may provoke new developments in related stereo-specific [1,3]-rearrangement and chiral Brønsted acid-catalyzed kinetic resolution, and the present protocol will find broad applications in synthetic and medicinal chemistry.

## Methods

**Materials**. Unless otherwise noted, materials were obtained commercially and used without further purification. All the solvents were treated according to general methods. Flash column chromatography was performed over silica gel (300–400 mesh). See Supplementary Methods for experimental details.

**General methods**. [1]H NMR spectra and carbon-13 nuclear magnetic resonance ([13]C NMR) spectra were recorded on a Bruker AV-400 spectrometer and a Bruker AV-500 spectrometer in chloroform-$d_3$. For [1]H NMR spectra, chemical shifts are reported in p.p.m. with the internal tetramethylsilane signal at 0.0 p.p.m. as a standard. For [13]C NMR spectra, chemical shifts are reported in p.p.m. with the internal chloroform signal at 77.0 p.p.m. as a standard. Infrared spectra were recorded on a Nicolet AVATER FTIR330 spectrometer as thin film and are reported in reciprocal centimeter (cm$^{-1}$). Mass spectra were recorded with Micromass QTOF2 Quadrupole/Time-of-Flight Tandem mass spectrometer using electron spray ionization. [1]H NMR, [13]C NMR, and HPLC spectra (for chiral compounds) are supplied for all compounds: see Supplementary Figs. 1–110. See Supplementary Methods for the characterization data of compounds not listed in this part.

**General procedure for the synthesis of 3-benzazocinones 2**. To a mixture of the ynamide **1** (0.20 mmol) in PhCl (3.75 mL) at room temperature, HOTf (0.001 mmol/0.25 mL) in 0.25 mL PhCl was added. Then, the reaction mixture was stirred at 80 °C and the progress of the reaction was monitored by thin layer chromatography (TLC). The reaction typically took 4 h. Upon completion, the mixture was concentrated and the residue was purified by chromatography on silica gel (eluent: hexanes/ethyl acetate) to afford the desired 3-benzazocinone **2**.

**General procedure for the synthesis of chiral 2-ent**. To a mixture of the yna-mide **1** (0.1 mmol) and 5 Å MS (60 mg) in Et$_2$O (2 mL) at room temperature, **Cat. 3** (0.02 mmol, 17.6 mg) was added during stiring. Then, the reaction mixture was stirred at 25 °C and the progress of the reaction was monitored by TLC. After the corresponding reaction time (6–32 h), Et$_3$N (0.03 mmol, 4.2 µL) and PhCl (1 mL) was added to the reaction mixure to quench the **Cat. 3**. The resulting reaction solution was stirred at 60 °C for another 24 h. The mixture was concentrated and the residue was purified by chromatography on silica gel (eluent: hexanes/ethyl acetate) to afford the desired chiral 3-benzazocinone **2**-ent.

## Data availability

Data for the crystal structures reported in this paper have been deposited at the Cambridge Crystallographic Data Center (CCDC) under the deposition numbers CCDC 1880379 (**2a**), 1880411 (**2ac**), 1880414 (**2ai**), and 1887308 (**2p**-ent). Copies of these data can be obtained free of charge via www.ccdc.cam.ac.uk/data_request/cif. All other data supporting the findings of this study, including experimental procedures and compound characterization, are available within the paper and its Supplementary Information files, or from the corresponding authors on request.

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

## Acknowledgements

We are grateful for financial support from the National Natural Science Foundation of China (21772161, 21622204, 21625204, 21702109, and 21890722), the President Research Funds from Xiamen University (20720180036), NFFTBS (No. J1310024), PCSIRT, Science and Technology Cooperation Program of Xiamen (3502Z20183015), the Fundamental Research Funds for the Central Universities (Nankai University: Nos. 63191515, 63191523, 63191321), and the Natural Science Foundation of Tianjin City (18JCYBJC21400). We also thank Professor Dr. Xianming Deng from Xiamen University (School of Life Sciences) for assistance with biological tests and Professor Dr. Nanfeng Zheng from Xiamen University (College of Chemistry and Chemical Engineering) for assistance with X-ray crystallographic analysis.

## Author contributions

B.Z., Y.-Q.Z., M.-Y.Y., Y.-B.C., and Y.L. performed experiments. K.Z. and Q.P. performed DFT calculations. Q.P., S.-F.Z., and Q.-L.Z. revised the paper. L.-W.Y. conceived and directed the project and wrote the paper. All authors discussed the results and commented on the manuscript.

## Additional information

**Competing interests:** The authors declare no competing interests.

