## [Peer Review File · Nature Communications]

Editorial Note: This manuscript has been previously reviewed at another journal that is not operating a transparent peer review scheme.

REVIEWERS' COMMENTS:

Reviewer #1 (Remarks to the Author):

I think this manuscript is appropriate for Nat. Comm. Although the text could be refined a bit, the authors have satisfactorily addressed my concerns. However, they might consider adding some references on concerted asynchronous reactions to the new section on the computational work (e.g., work from Rzepa, Hess, Tantillo, Dewar, Houk, Cremer) and it would be beneficial to see the results of IRC calculations on their transition states, since these would show that there is not a second transition state.

1. Response to comments (reviewer 1):

I think this manuscript is appropriate for Nat. Comm. Although the text could be refined a bit, the authors have satisfactorily addressed my concerns. However, they might consider adding some references on concerted asynchronous reactions to the new section on the computational work (e.g., work from Rzepa, Hess, Tantillo, Dewar, Houk, Cremer) and it would be beneficial to see the results of IRC calculations on their transition states, since these would show that there is not a second transition state.

- We first thank the reviewer for her/his kind recommendation and suggestions.
- The following literatures have been added as Refs. 66-70 according to the reviewer's suggestion:

2. For concerted asynchronous reactions, see references ⁶⁶⁻⁷⁰. Agrafiotis D. K. & Rzepa H. S. Dihydrogen transfer reactions. An SCF-MO study of the relative energies of the concerted and stepwise pathways. *J. Chem. Soc., Chem. Commun.* 902-904 (1987).
3. Tantillo D. J. Recent excursions to the borderlands between the realms of concerted and stepwise: carbocation cascades in natural products biosynthesis. *J. Phys. Org. Chem.* **21**, 561–570 (2008).
4. Hess Jr., B. A. & Smentek L. The concerted nature of the cyclization of squalene oxide to the Protosterol cation. *Angew. Chem. Int. Ed.* **52**, 11029 –11033 (2013).
5. Pham, H. V. & Houk, K. N. Diels–alder reactions of allene with benzene and butadiene: Concerted, stepwise, and ambimodal transition states. *J. Org. Chem.* **79**, 8968–8976 (2014).
6. Mackey, J. L., Yang, Z.-Y & Houk, K. N. Dynamically concerted and stepwise trajectories of the cope rearrangement of 1,5-hexadiene. *Chem. Phys. Lett.* **683**, 253–257 (2017).